# High-Performance Broadband Bistatic Piezoelectric Composite Array for Application in Ship Wake Detection

**DOI:** 10.3390/ma16062199

**Published:** 2023-03-09

**Authors:** Fenghua Tian, Wenqiang Tian, Yiming Liu, Ruilei Ma, Yongquan Ding, Bao’an Hao

**Affiliations:** Xi’an Precision Machinery Research Institute, Xi’an 710077, China

**Keywords:** 1–3 composite array, broadband, bistatic array, ship wake, array arrangement method

## Abstract

In the far-field wake of a ship, the intensity of the scattering of bubbles is relatively weak. In addition, the wake is relatively thin, and the hole phenomenon is prominent. Thus, it is difficult to detect the wake at a long distance. On this basis, this paper studies a broadband 1–3 high-performance composite transceiver sub-array for the improved detection of a ship’s far wake flow field. The content includes frequency characteristics, transmission performance, power tolerance, the beam width of the transmitting array, and the frequency characteristics, reception performance, and beam width of the receiving array. The frequency bandwidth of the transmission array developed in this paper can reach a value of 180 kHz (the center frequency is 390 kHz). The maximum sound source level can reach a value of 228 dB. In the same frequency band, the sensitivity of the receiving array can reach a value of 184 dB, and the fluctuation is less than 5 dB. Compared with the narrowband 1–3 composite array of the same size, the acoustic performance of this sub-array has obvious advantages. Finally, to improve the effective sound path (before the first interface reflection of the sound wave) of the emitted sound wave in the ship’s far-field wake, combined with the speed of the moving carrier and the wide-band detection method of the ship’s wake, the configuration method of the detection array for the width and direction of the ship’s far-field wake is proposed. The results of this research have an important reference value for the research on broadband 1–3 high-performance composite arrays and their application in the far-field wake detection of ships.

## 1. Introduction

A 1–3 piezoelectric composite is a functional material composed of one-dimensionally connected piezoelectric material columns periodically arranged in a three-dimensionally connected polymer according to a certain volume ratio and spatial geometric size. Compared with single-type piezoelectric ceramic materials, a 1–3 piezoelectric composite usually has the following advantages:

(1) A 1–3 piezoelectric composite is easy to match with the water medium and can reduce the noise interference of the receiving channel because of its lower acoustic impedance Z. (2) The receiving voltage sensitivity of the acoustic array can usually be improved by using a 1–3 piezoelectric composite to decrease the dielectric constant $\varepsilon_{r}$ and increase the hydrostatic piezoelectric constant $g_{h}$. (3) The vibration mode of a 1–3 piezoelectric composite is purer than that of a single-phase piezoelectric ceramic, which can eliminate clutter interference because the radial electromechanical coupling coefficient $k_{p}$ of a 1–3 piezoelectric composite is small while its thickness electromechanical coupling coefficient $k_{t}$ is large. When used for emission, it can improve the acoustic electrical conversion efficiency of specific frequency bands [1]. (4) The low mechanical quality factor $Q_{m}$ of a 1–3 piezoelectric composite can expand the bandwidth of the acoustic array [2,3].

Ship wake can be reduced but not eliminated. The technology for detecting the near wake area is relatively mature, but there is little research on the technology for detecting the far wake area, where the length of the ship’s wake exceeds 300 $V_{k}$ (the product of ship speed and time). The thickness of the wake in the far wake field is greatly reduced, and there are electron holes, which are difficult to achieve by conventional narrowband detection methods. In this paper, on the basis of the technology for detecting a ship’s far wake flow field under the high-speed moving carrier and its wake characteristics, we have studied and tested a broadband 1–3 high-performance transceiver–transmitter composite array for detecting the far wake flow field. In addition, we have proposed and studied the configuration method of the wideband, high-performance transmitter and receiver split array when the far wake of a ship is 4 m thick and 50 m wide and the speed of the array carrier is 50 knots.

## 2. Study on the Characteristics of the Far Wake Flow Field

The far wake flow field of a ship with a wake length of more than 300 $V_{k}$ is characterized by a constant wake width and small thickness. Detecting the far wake flow field mainly involves calculating, among other parameters, the critical radius of bubbles, the scattering intensity, and the bubble relative number density.

The bubble radius of the far wake flow field of a ship is near the critical radius, which is about 30~70 μm [4]. The scattering ability of wake bubbles is related to bubble size (a), initial depth ($H_{0}$), frequency (f), and other factors [5,6]. The farther the wake field, the smaller the bubble radius, the higher the detection resolution, and the higher the optimal detection frequency [7,8]. The scattering ability is expressed by the logarithm of the ratio of the scattering cross section to the cross-sectional area, as shown in Figure 1. The change in the bubble relative number density with time is shown in Figure 2. The main frequency band of the high-performance acoustic array selected in this study is 330~480 kHz. The relative number density of bubbles is about 25% [9,10].

The echo level of ship wake can be expressed as:(1)EL=SL−40logr+W+10logL
(2)L=Φr
(3)$$10\log F=10\log\frac{l}{2pa}+6.9$$
where EL is for the echo level with a wake intensity of W, SL is for the emission sound source level, 40×logr is the spherical attenuation, r is for the detection range, W is the wake intensity (generally, −15 dB is taken for ships, and the attenuation with the wake detection length is about 1 dB/min), *L* stands for the length of the wake echo, Φ is the equivalent plane angular beam width, and a is the radius of the transmitting plane circular array [11,12].

According to the passive sonar equation, the technical parameters of a 1–3 high-performance bistatic acoustic array can be calculated as follows:(4)EL−TL=NL−DI+DT
(5)$$DI=10\log{\left(\frac{\pi D}{\lambda }\right)}^{2}$$
(6)$$TL=20\log(r)+\alpha r\times 10^{-3}$$
(7)$$\alpha =\frac{0.1f^{2}}{1+f^{2}}+\frac{40f^{2}}{4.1+f^{2}}+2.75\times 10^{-4}\times f^{2}+0.003$$
where TL is for propagation loss, NL is for the self-noise level, DI is for the receiver directivity index, D is for the receiver array diameter, DT stands for the detection domain, α is for the absorption coefficient, and f is for frequency.

In this study, the one-way detection distance of the far wake flow field studied is r = 15 m. According to the characteristics of the far wake flow field of the ship, the echo intensity of the wake, and other research results, the source level of the transmitting array is calculated to be greater than 221 dB, and the sensitivity of the receiving array is greater than −195 dB. To reduce the impact of interface reflection on the receiving performance, the transmitting beam width is 3.5 ± 1° and the receiving beam width is 11 ± 1°.

## 3. Theoretical Study on a High-Performance Bistatic Acoustic Array

On the basis of the results of far wake flow field characteristics and wake detection calculation, we studied the circular 1–3 high-performance acoustic emission array, the octagonal 1–3 high-performance acoustic emission array, and the circular 1–3 high-sensitivity receiving array, and on this basis, we used the matched layer method to study the broadband acoustic performance of transmitting and receiving arrays.

In this study, the base material of 1–3 composite materials used for emission was PZT-4-modified piezoelectric ceramic, and the filling material was epoxy resin (Table 1 presents the material parameters). The substrate of the 1–3 composite material used for receiving was PZT-5-modified piezoelectric ceramic because of its high voltage and high dielectric constant. The filling material was the same as that of the transmitting transducer.

### 3.1. Circular Transmitting Array

According to the requirement of the transmitting beam width, the array aperture was 77 mm, the array element spacing (d) was 1.92 mm, the gap between array elements was 0.3 mm, and the thickness was 4.2 mm, as shown in Figure 3.

To calculate the corresponding frequency of the transmitting voltage of the circular array, we used the finite element method and harmonic response analysis [13,14]. The finite element model is shown in Figure 4. On the basis of the theory of an arbitrarily shaped piston array with a point source, we simulated the directivity of the circular launch array with Matlab software [15].

For a discrete array of uniform transducers, the three-dimensional directivity function with the working angle frequency of ω and the number of elements of N is: (8)$$D(\alpha ,\theta ,\alpha_{0},\theta_{0},\omega )=\frac{\left|\sum_{i=1}^{N}A_{i}e^{-j\Delta \varphi_{i}}\right|}{\left|\frac{1}{N}\sum_{i=1}^{N}A_{i}\right|}$$

In this equation, the coordinate origin O is taken as the reference point, where $A_{i}$ is the amplitude of the sound pressure generated by the ith element of the array at far-field P points. The phase difference $\Delta \varphi_{i}$ between the sound pressure of the sound wave radiated from the ith array element of the array in any direction at the far-field P point and the sound wave radiated in the reference direction of the coordinate origin is:(9)$$\begin{array}{ll} \Delta \varphi_{i} & =\varphi_{i}-\varphi_{0}=\frac{\omega }{c}(\xi_{i}-\xi_{i0})\\ & =\frac{\omega }{c}[x_{i}(\sin\theta \cos\alpha -\sin\theta_{0}\cos\alpha_{0})\\ & +(y_{i}\sin\theta \sin\alpha -\sin\theta_{0}\sin\alpha_{0})+z_{i}(\cos\theta -\cos\theta_{0})] \end{array}$$
where c is the velocity of sound in water; $x_{i}$, $y_{i}$, and $z_{i}$ are the position coordinates of the array element; and the main beam is in the $\theta_{0}=0^{\circ }$ direction.

Figure 5 and Figure 6 shows the simulation results.

As per Figure 5 and Figure 6, the corresponding level of the maximum transmitting voltage can reach a value of 190 dB, and the −6 dB bandwidth is 365~430 kHz. The transmitted beam of the array has an axisymmetric and central symmetric pattern, with a −3 dB beam width of 3.5°, a main side lobe ratio of 17 dB, small side lobes, and more concentrated transmitting energy.

Since the radius of the bubbles in the far wake field varies greatly, the corresponding optimal detection frequency is also higher. To improve the detection efficiency and obtain more wake information based on the results of the narrowband 1–3 composite array, we studied the longitudinal dual-resonant broadband composite launch array by using the matching layer technology. Figure 7 displays the emission characteristics.

On adding the matching layer, the maximum transmitting voltage response of the broadband acoustic array can reach a value of 188 dB (Figure 7). Within the frequency band range of 350 kHz and 490 kHz, the response fluctuation is less than 6 dB, and the transmitting frequency band width is significantly broadened.

### 3.2. Octagonal Transmitting Array

In order to suppress the influence of the side lobe on the reception performance, improve the main side lobe ratio, and increase the wake reverberation sound scattering in specific directions, we designed the octagonal composite array when the vertical beam was 3.5°. The surface area of the array was equivalent to that of the circular emission array; the array spacing was the same; 48 arrays were evenly spaced in the vertical L direction, and 8 arrays were evenly spaced in the horizontal d direction; there were a total of 1176 arrays, and the radiation mask had a 90° rotational symmetry. The horizontal directivity was consistent with the vertical directivity, and its structure is shown in Figure 8.

In the figure, the short side of the octagon array is 15 mm long, the long side is 75 mm long, and the vertical and horizontal lengths are 91 mm each.

Similarly, the beam of the octagonal composite array was calculated according to Equation (8), and the result is shown in Figure 9.

It can be seen from Figure 8 that when the frequency is 390 kHz, the main beam width of the array in the horizontal and vertical directions is still 3.5°, but the level of the first side lobe is reduced to −27 dB. The side lobe level is higher in the positive and negative 45° directions, and the first side lobe level is −15 dB.

Compared with the circular composite array, the level of the first side lobe is reduced by 10 dB in the horizontal and vertical directions, and the side lobe is effectively suppressed. In the positive and negative 45° directions, the side lobe level is increased by 2 dB, and the sound scattering intensity is increased in the positive and negative 45° directions.

### 3.3. Circular Receiving Array

The circular receiving array we studied used 1–3 composite arrays, the piezoelectric matrix used PZT-5-modified materials, and the filler was the same as the transmitting array. To ensure a better match between the received beam and the transmitted beam when the moving carrier’s speed was 50 knots and the frequency was 390 kHz, the designed beam width of the array was 11.5°, the array aperture was 20 mm, the array was a 1.6 mm × 1.6 mm column, the array spacing was *d* = 1.9 mm, the receiving frequency band was consistent with the transmitting array, and the thickness was 4.3 mm.

The receiving sensitivity (*MeL*), which is the ratio of the sound pressure value received by the hydrophones to the voltage generated by the 1–3 hydrophones of the circular receiving array, satisfied the following equation:(10)$$MeL=g_{h}t$$

In the above formula, $g_{h}$ is a constant, and t is the thickness of 1–3 composites.
(11)$$g_{h}=(d_{33}+2d_{31})/\varepsilon^{T}{}_{33}$$

In the above formula, the piezoelectric constants $d_{33}$ and $d_{31}$ the dielectric constant $\varepsilon^{T}{}_{33}$ are the piezoelectric charge constants of the composite and are related to the piezoelectric material parameters and the volume proportion in the composite. The specific calculation method is provided in [1], and the highest sensitivity of the receiving array can reach a value of −196 dB through calculation.

According to the calculation formula of any array, we simulated and calculated the beam of this array, and the frequency was 390 kHz. The results are shown in Figure 9.

As can be seen from Figure 10, when the frequency of this circular composite receiving array is 390 kHz, its beam width is 11.5°, and the value of the first side lobe is −18 dB.

## 4. Experimental Study on a High-Performance Bistatic Acoustic Array

In this study, we developed three types of 1–3 acoustic arrays: a circular transmitting array, an octagonal transmitting array, and a circular receiving array. Each type includes narrowband and broadband types [16,17,18].

### 4.1. Fabrication and Testing of a Circular Transmitting Array

According to the theory and simulation calculation results, with the help of special tooling and molds, we created a broadband 1–3 launch array of matching layers and poured a matching layer that was 10 mm thick. According to the acoustic performance requirements, the thickness of the matching layer was finally determined to be 4.7 mm. To facilitate comparison, the narrowband 1–3 emission arrays without the matching layer were fabricated together, as shown in Figure 11.

In the anechoic tank, we measured the transmitting beam and the transmitting performance with the help of an automatic measurement system belonging to the third-level underwater acoustic metering station. The theoretical calculation was compared with the measured results. The results are shown in Figure 12 and Figure 13.

In Figure 12, the main beam width at 390 kHz is 3.1°, and the main side lobe ratio of the beam is 15 dB. The theoretical design value is consistent with the measured value. It can be seen from Figure 13 that, at the resonant frequency, the maximum value of the transmitted voltage response of the circular transmitting array of the unperfused matching layer is 192 dB and the −6 dB bandwidth of the circular transmitting array of the perfusion matching layer is 350~420 kHz. The maximum value of the transmitting voltage response level at the first resonant frequency point (360 kHz) is 189 dB, and the transmitting voltage response at the second resonant frequency point (480 kHz) is 184 dB. The −6 dB bandwidth is 320~500 kHz. The theoretical design value is basically consistent with the measured value. For the circular transmitting array of the perfusion matching layer, within the −6 dB bandwidth, the minimum impedance is 28 Ω, the minimum transmitting voltage is 183 dB, the input $V_{pp}$ = 260 V voltage, the minimum source level can reach a value of 222 dB, and the maximum power consumption is 302 W.

### 4.2. Fabrication and Testing of an Octagonal Transmitting Array

According to the theoretical design results of the octagonal array, we designed and developed narrowband and matching layer broadband octagonal emission arrays, as shown in Figure 14.

In the anechoic tank, we measured the transmitting performance and the transmitting beam performance with the help of an automatic measurement system and compared the theoretical calculation with the measured results. The results are shown in Figure 15 and Figure 16.

It can be seen from Figure 15 and Figure 16 that the main beam width at f = 390 kHz is 3.2°, the first side lobe level at 0° and 90° is −26.8 dB, and the first side lobe level at 45° is −13 dB. The theoretical design value is in good agreement with the measured value. It can be seen from Figure 16 that the resonant frequency of the transmitting array of the unperfused matching layer is 390 kHz and the corresponding maximum transmitting voltage response is 193 dB, but the −6 dB bandwidth is narrow. The first-order resonant frequency of the matching layer transmitting array is 375 kHz, the transmitting voltage response is 187 dB, the second-order resonant frequency is 495 kHz, and the transmitting voltage response is 181 dB. The matching layer length of the transmitting array is not enough, leading to large fluctuations in the frequency band, but the broadband characteristics are obvious.

### 4.3. Fabrication and Testing of a Circular Receiving Array

According to the theoretical design results of the circular receiving array, we designed and developed narrowband and matching layer broadband circular receiving arrays, as shown in Figure 17.

In the anechoic tank, we measured the receiving performance with the aid of an automatic measurement system. To reduce the impact of the distributed capacitance of the test cable, we configured a 6 dB amplification module close to the receiving array. Then, we compared the theoretical calculation with the measured results. The results are shown in Figure 18 and Figure 19, respectively.

It can be seen from Figure 18 that when f = 390 kHz, the −3 dB receiving beam width of the circular receiving array is 11.5° and the first side lobe level is −21 dB. The theoretical design value is in good agreement with the measured value. It can be seen from Figure 19 that the maximum receiving sensitivity of the broadband 1–3 circular receiving array based on matching layer technology can reach −184 dB; its receiving sensitivity is greater than −190 dB; its fluctuation is less than 5 dB in the frequency band of 330~500 kHz; and compared with a narrowband circular receiving array, it has excellent broadband characteristics and receiving performance.

## 5. Method of Configuring a Bistatic Array

There are three kinds of reverberation in a ship’s wake: volume reverberation, interface reverberation, and wake reverberation. Ship wake detection involves detecting wake volume reverberation under interface reverberation. In the wake area, the conventional detection method is usually the amplitude (energy) detection method. The corresponding algorithm is relatively mature and reliable, but the precondition is that the ship wake should have sufficient thickness. The wake field studied in this paper is the far wake area of a ship with a distance of more than 300 Vk. The wake is 4 m thick and 50 m wide; the array carrier speed is 50 knots; and the bubble relative density is about 25%. In such conditions, it will be more difficult to obtain valuable information about the direction and thickness of ship’s wake.

As an effective method of detecting a ship’s far wake flow field, we propose the configuration model of a transmitting array and receiving array, as shown in Figure 20.

In Figure 20, the acoustic axes of the transmitting array and the receiving array are in the XOY plane, and the array carrier moves along the positive direction of the *X* axis. To reduce the impact of flow noise and radiation noise on the receiving array, the receiving array is in the front and the transmitting array is in the back. The included angle between the receiving array and the *Y* axis is $\alpha_{1}$, and the distance to the central axis of the array carrier is H1. The included angle between the transmitting array and the *Y* axis is $\alpha_{2}$, the distance to the central axis of the array carrier is H2, and the central distance between the transmitting array and the receiving array is D.

The optimal configuration of the transmitting array and the receiving array must meet two conditions: (1) in wake reverberation, the transmitting beam must be all in the receiving beam and the two should belong to the inclusion relationship and (2) when the array carrier moves at a high speed, when the transmitting beam of the transmitting array reaches the interface for primary reflection, its beam should be located at the center of the receiving array beam. The beam of the transmitting array developed in this paper is 3.5 ± 1°, the receiving beam is 11 ± 1°, and the speed of the array carrier is 50 knots. The theoretical calculation results are shown in Figure 21. The configuration of the transmitter receiver array is shown in Table 2.

The transmitting array emits sound waves in the wake reverberation of the ship. Before the sound wave arrives at the interface (interface reverberation), the sign of effective detection is that the effective signal received by the transmitting sound wave within the effective sound path of the wake reverberation is different from the volume reverberation received by the receiving array. The effective sound path is related to the working depth of the array carrier, the wake width and wake depth of the ship’s far wake field, and the rotation angle (roll angle) of the plane where the transmitter receiver array is located along the axis of the array carrier.

Assuming that the array carrier is sailing at the wake centerline, the best condition for the maximum sound path of the acoustic wave emitted by its transmitting array in the wake reverberation is that the emitted acoustic wave penetrates along the edge of the ship’s wake width. Figure 22 shows the theoretical calculation results, and Table 3 presents the optimal wake detection technical parameters. Line A–C is the acoustic-path of the sound wave in the wake. 

It can be seen from Table 3 that the effective detection time along the direction (lateral) of the ship’s wake width is more than three times longer than that in the 0° direction before the interface reflects once. The most effective sound path for wake field detection can also be obtained by changing (1) the width and thickness of the ship wake and (2) the speed and depth of the array carrier.

## 6. Conclusions

In this study, we studied the wake characteristics, such as the density of the bubbles, wake width, and wake depth, and the relationship between the bubble scattering ability and frequency. The results show that, (1) in combination with the high-frequency wake detection signal, fast signal attenuation in water, difficulty of interception, and the best detection bandwidth, a broadband high-performance bistatic separated acoustic array of 1–3 matching layers is suitable for far wake flow field detection; and (2) in the frequency range of 330~480 kHz, the minimum source level of the transmitting array can reach 222 dB (power consumption: 302 W) and the receiving sensitivity of the receiving array is greater than −190 dB. Using the high-performance broadband transmitter and receiver arrays developed in this study, we proposed a configuration method for the transmitter and receiver arrays corresponding to the detection of the ship’s far wake flow field. Finally, we optimized the configuration of the transmitter receiver array under the specific conditions of the far wake flow field. These results may benefit the development of broadband 1–3 high-performance composite arrays and their applications in the far-field wake detection of ships.

## Figures and Tables

**Figure 1 materials-16-02199-f001:**
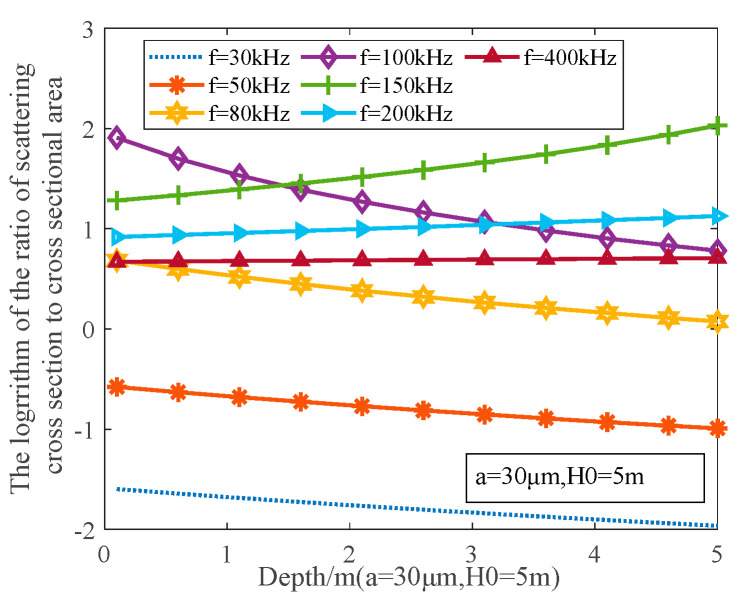
Scattering Ability of Wake Bubbles at Different Frequencies.

**Figure 2 materials-16-02199-f002:**
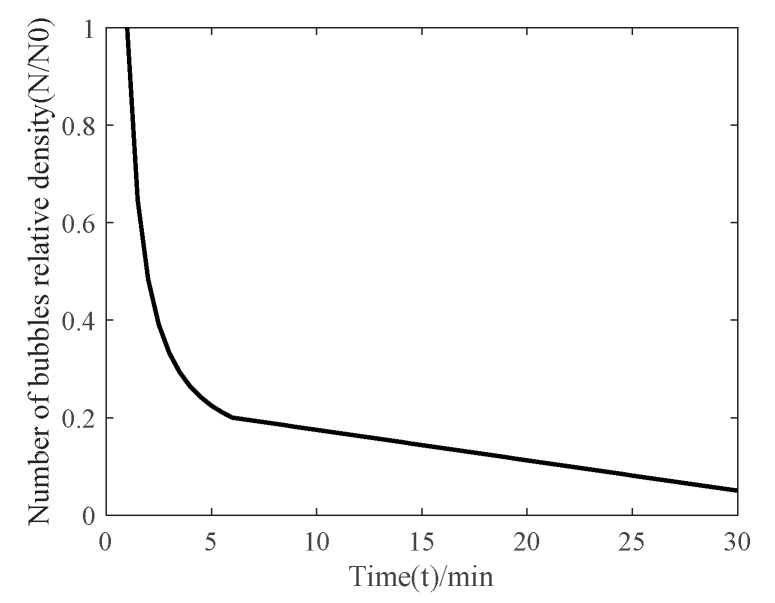
Variation Curve of Bubble Relative Number Density.

**Figure 3 materials-16-02199-f003:**
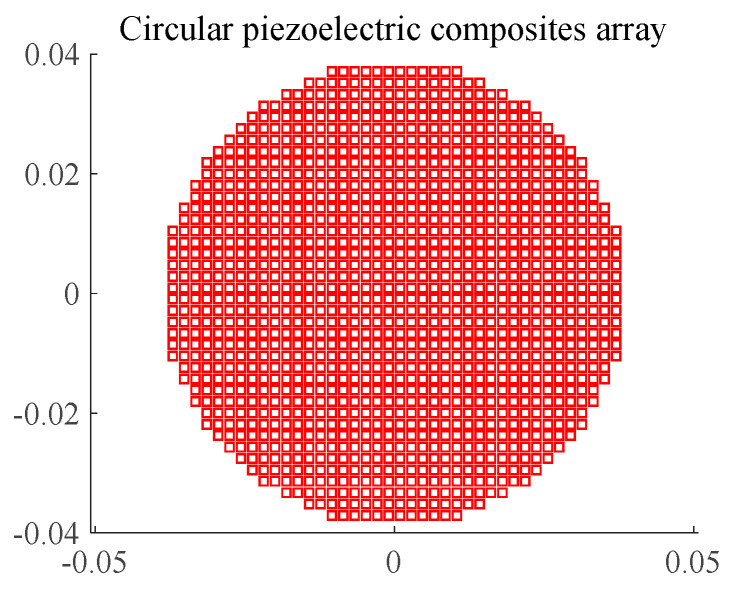
Circular Transmitting Array.

**Figure 4 materials-16-02199-f004:**
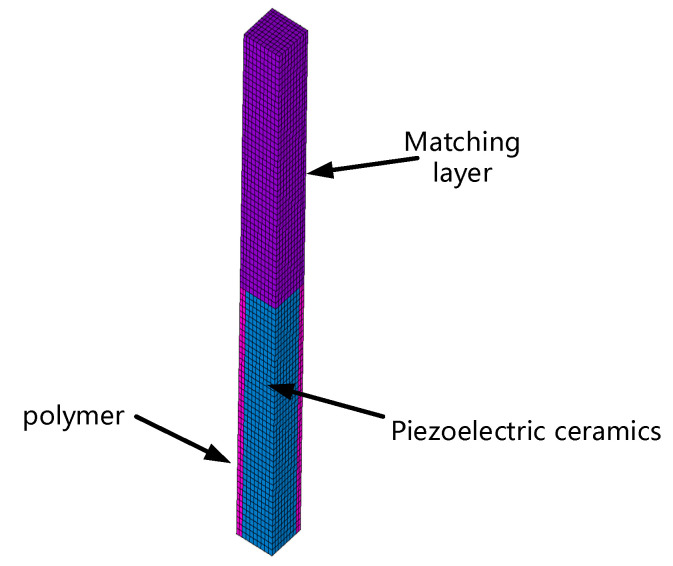
1–3 Finite Element Model of Piezoelectric Composites.

**Figure 5 materials-16-02199-f005:**
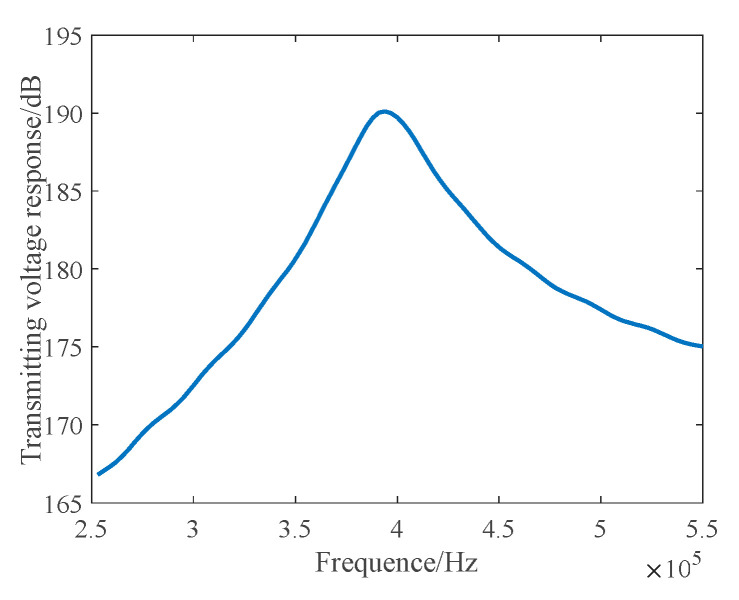
Transmitting Voltage Response–Frequency Curve.

**Figure 6 materials-16-02199-f006:**
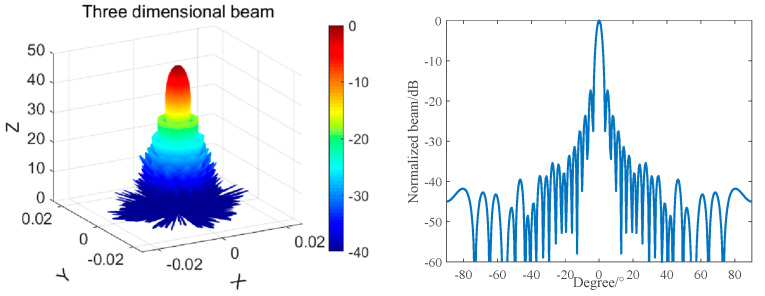
Beam of the Circular Transmitting Array with f = 390 kHz.

**Figure 7 materials-16-02199-f007:**
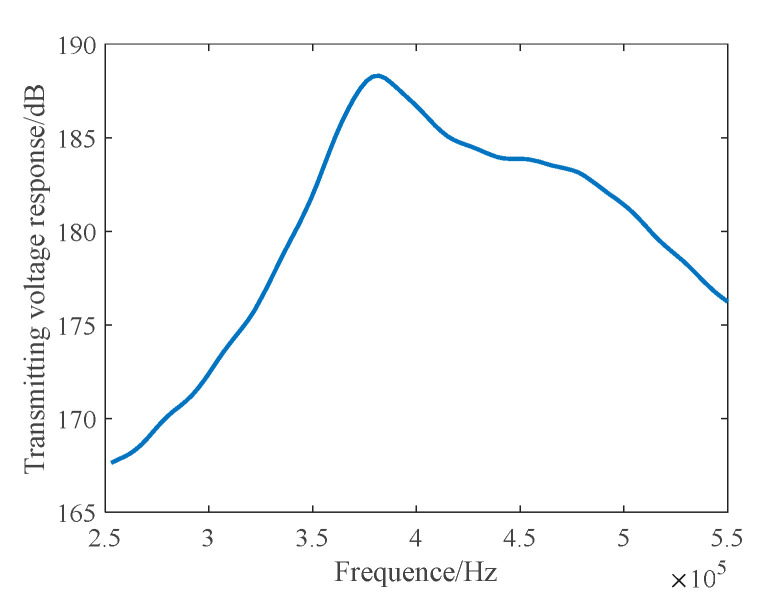
Transmitting Voltage Response–Frequency Curve of a Broadband Composite Array in the Matching Layer.

**Figure 8 materials-16-02199-f008:**
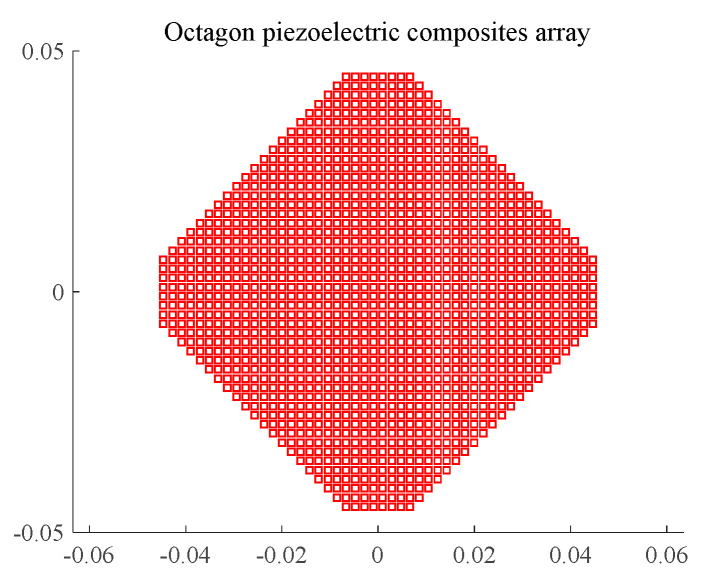
Octagonal 90° Rotating Symmetric Composite Transmitting Array Structure.

**Figure 9 materials-16-02199-f009:**
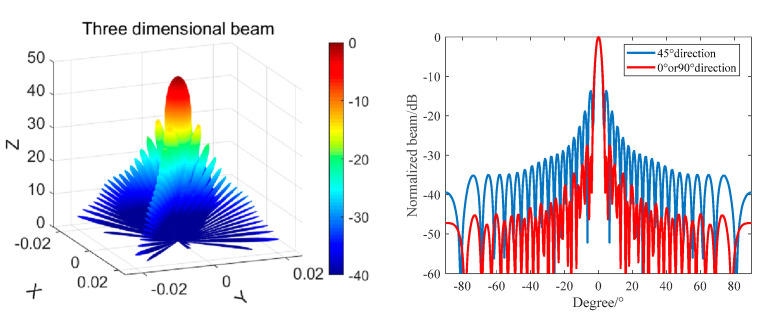
An octagonal rotationally symmetric composite array beam with F = 390 kHz.

**Figure 10 materials-16-02199-f010:**
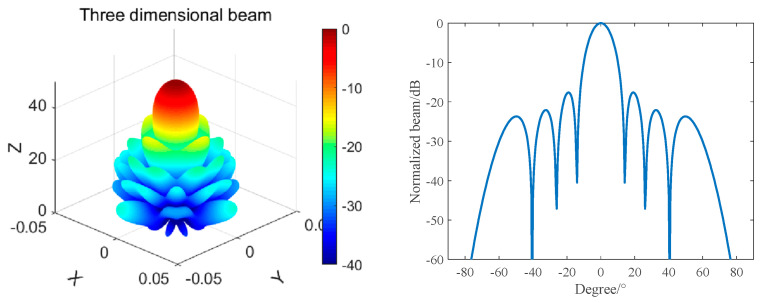
Beam of the 390 kHz Circular Composite Receiving Array.

**Figure 11 materials-16-02199-f011:**
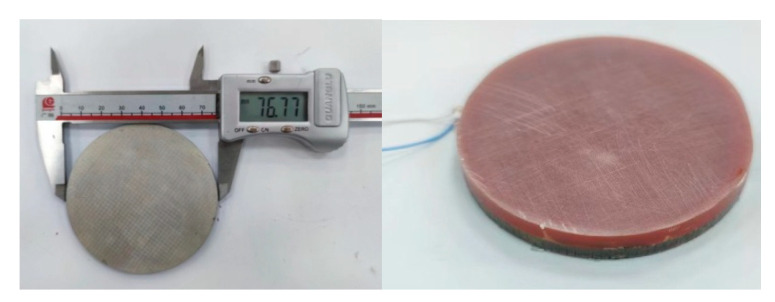
Circular Transmitting Array.

**Figure 12 materials-16-02199-f012:**
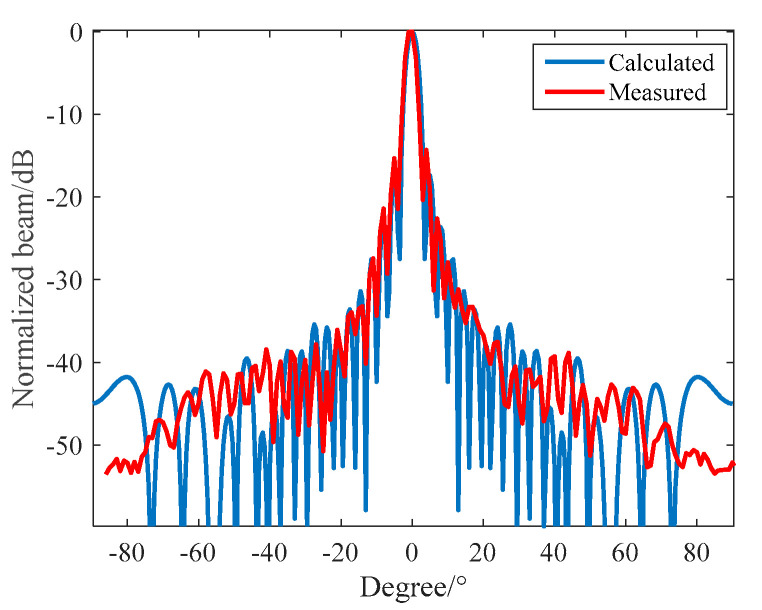
Directionality of the 390 kHz Circular Transmitting Array.

**Figure 13 materials-16-02199-f013:**
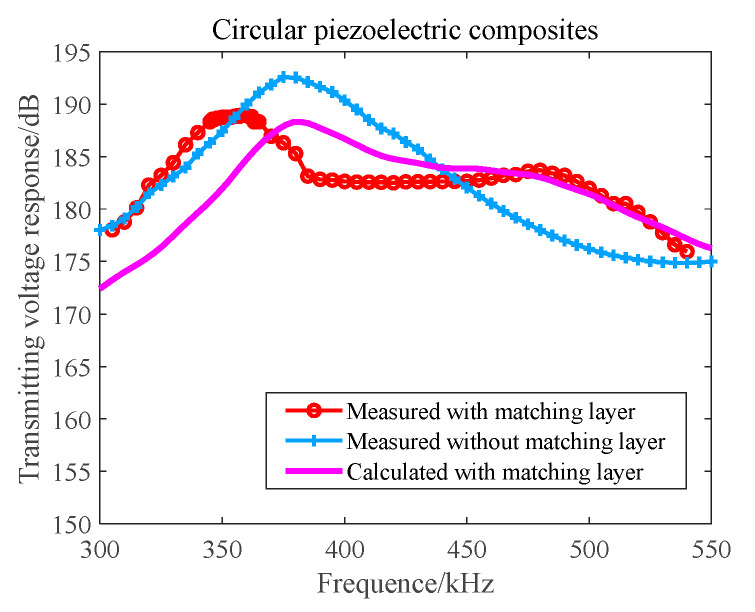
Transmitting Voltage Response–Frequency Curve of the Circular Transmitting Array.

**Figure 14 materials-16-02199-f014:**
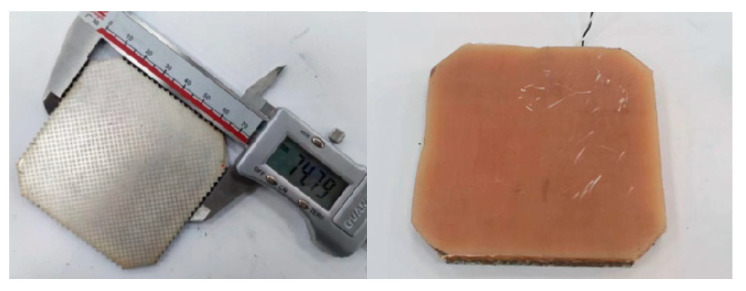
Narrowband (**Left**) And Broadband (**Right**) Octagonal Emission Arrays.

**Figure 15 materials-16-02199-f015:**
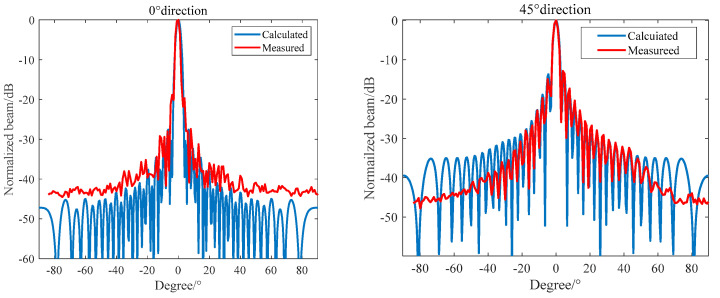
0° and 45° Directivity of a 390 kHz Octagonal Transmitting Array.

**Figure 16 materials-16-02199-f016:**
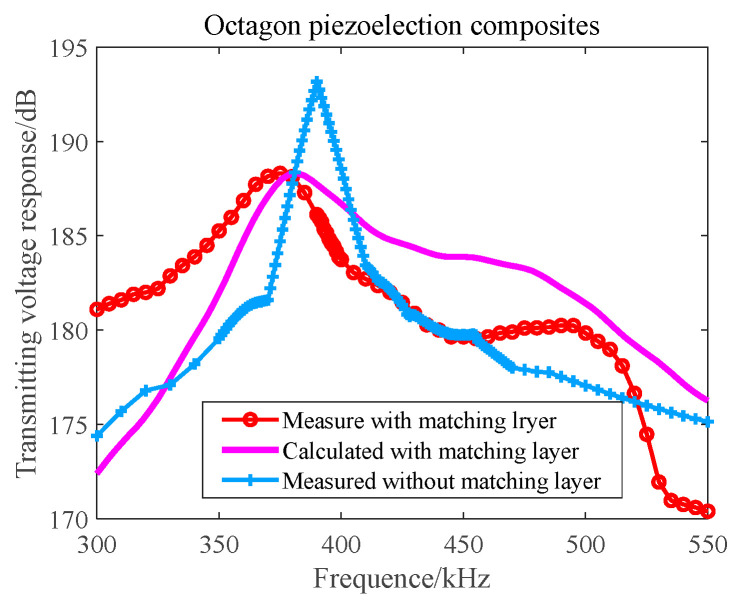
Transmitting Voltage Response–Frequency Curve of an Octagonal Transmitting Array.

**Figure 17 materials-16-02199-f017:**
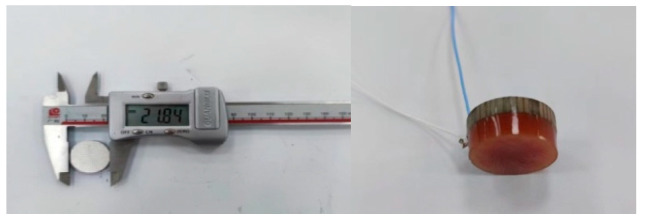
Narrowband (**Left**) And Broadband (**Right**) Receiving Arrays.

**Figure 18 materials-16-02199-f018:**
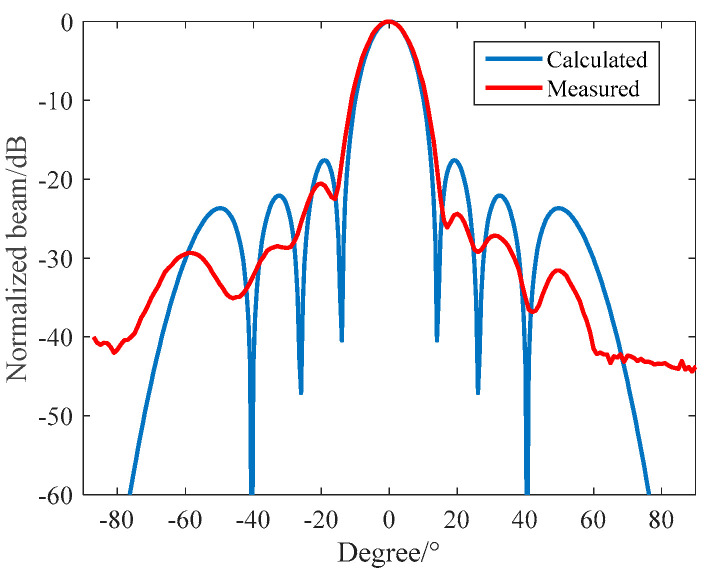
Directionality of a 390 kHz Circular Receiving Array.

**Figure 19 materials-16-02199-f019:**
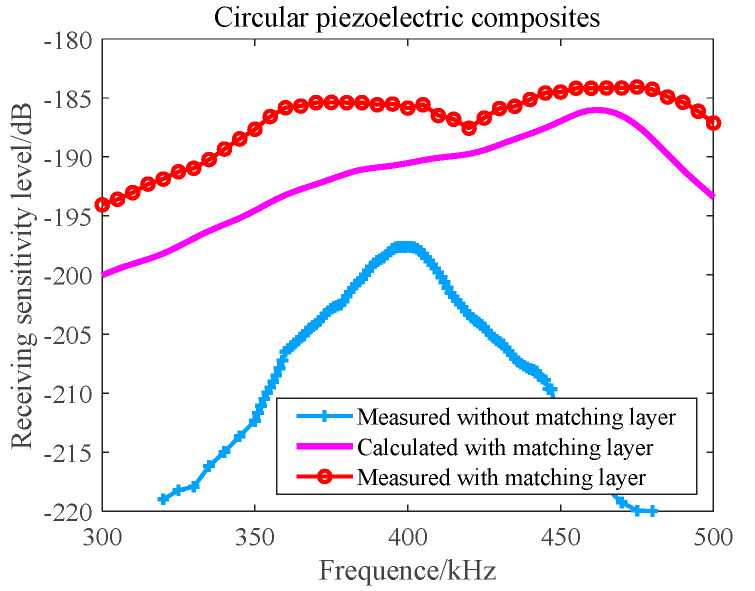
Receiving Sensitivity–Frequency Curve of a Circular Receiving Array.

**Figure 20 materials-16-02199-f020:**
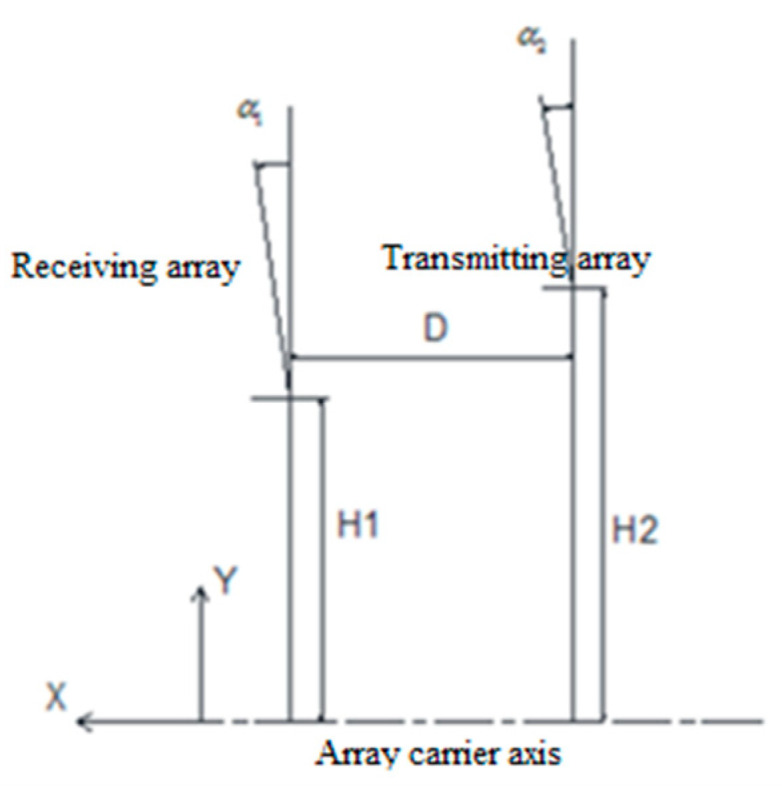
Configuration Model of Bistatic Array.

**Figure 21 materials-16-02199-f021:**
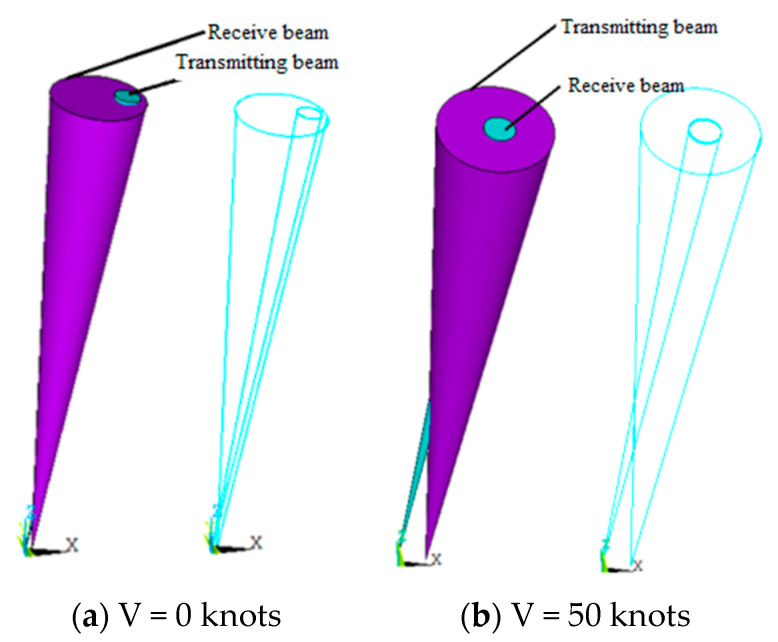
Beam Configuration of Transmitting and Receiving Arrays.

**Figure 22 materials-16-02199-f022:**
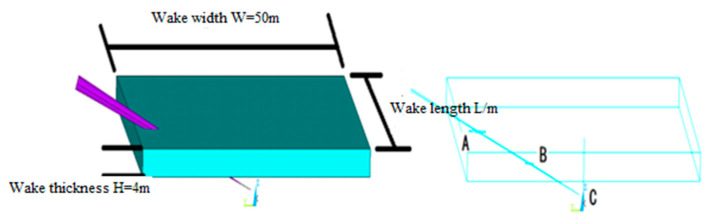
Schematic Diagram of The Position of The Transmitting Array in The Wake When γ=70°.

**Table 1 materials-16-02199-t001:** Material Parameters of the Epoxy Resin Filler.

Elastic Modulus (E_x_/N × m^−2^)	Hardness (HB)	Poisson’s Ratio, σ	Density, ρ (Kg × m^−3^)
3.55 × 10^9^	83	0.41	1900

**Table 2 materials-16-02199-t002:** Position Parameters of a Bistatic Array.

H1/mm	H2/mm	D/mm	$\alpha_{1}$/°	$\alpha_{2}$/°	γ/°
130.8	174.94	114.6	8	11	0/70

**Table 3 materials-16-02199-t003:** Technical Parameters of Wake Detection.

γ/°	LAC/m	LAB/m	T/ms
0	8	4	2.7
70	23.34	12.7	8.47

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
