# Peer review of "High-Performance Broadband Bistatic Piezoelectric Composite Array for Application in Ship Wake Detection"

_materials, 2023, doi:10.3390/ma16062199_

Round 1
Reviewer 1 Report
Paper on 1-3 piezocomposites. I have the following comments:
The English and presentation can be improved.
Images - small text and small figures - this comment applies to most figures. Figure 8 is very small.
Table 1 - Check Young’s modulus of epoxy - often ~1-3GPa but it is much lower in Table 1?
Space after no. and units in ALL cases, e.g. 390 kHz, not 390kHz
Limited details of the finite element model, please provide more details.
Why use PZT-5 material? Explain.
Define what is MeL (sensitivity?) - equation 9 Can be mistaken as M*e*L or MeL?
Equation 10 is not right , it should be divided by ABSOLUTE permittivity and not divided by RELATIVE permittivity (dielectric constant).
Figure 10 - images are not good as it is very difficult to see the ceramic pillars of the 1-3 structures - better and higher magnification figures are needed.
Porous materials are another potential route and maybe of interest:
Ultrasonic Transducers Made From Freeze-Cast Porous Piezoceramics
Z Rymansaib et al IEEE Transactions on Ultrasonics, Ferroelectrics, and Frequency Control 69 1 2022
Porous ferroelectric materials for energy technologies: Current status and future perspectives M Yan et al, Energy & Environmental Science 14 (12), 6158-6190 18 2021
Reviewer 2 Report
"Review comments" Comments on the manuscript " High Performance Broadband Bistatic Piezoelectric Composite 2 Array for Applications in Ship Wake Detection ". In this paper, the various application effects of different 1-3 including frequency characteristics were investigated. The samples are well characterized, and the data is explained well. As a whole, the topic is interesting and is suitable for publication in Materials. I have the following comments, which may be considered before accepting the manuscript for publication.- In Figure 1, the X-axis and Y-axis legends are not in English. This should be corrected in English. Apart from that, all figures must have high resolution and be clear.
- Before the manuscript is published, English sentences should be revised to make them more understandable and clear. Overall English is very difficult to understand, and there are some sentences that are not grammatically correct. English needs to be corrected.
- In the field of piezoelectric research, many studies to replace lead have been conducted for a long time. What is the author's opinion on the development of arrays using lead-free piezoelectric materials?
Reviewer 3 Report
Here are my concerns:
- [Introduction] Should be more organized and informative. The last paragraph of the Introduction must be modified and updated with the novelty, scientific soundness, technical issues, and methods used, as well as quantified results of the proposed work. Current content does not have any adequate information.
- All figures quality needs /to be improved. Please check the figure 1 caption (language).
- Please check the figure 8. The simulation result is not clearly visible. Figures 11, 14, and 17, please add Y-axes caption.
- The authors should add the performance comparison with previously reported work.
5. The authors should include the stability and durability test results for real devices.
Round 2
Reviewer 3 Report
No more comments.